# Strengthening the Evidence for a Causal Link between Type 2 Diabetes Mellitus and Pancreatic Cancer: Insights from Two-Sample and Multivariable Mendelian Randomization

**DOI:** 10.3390/ijms25094615

**Published:** 2024-04-23

**Authors:** Te-Min Ke, Artitaya Lophatananon, Kenneth R. Muir

**Affiliations:** Division of Population Health, Health Services Research and Primary Care, School of Health Sciences, Faculty of Biology, Medicine and Health, The University of Manchester, Manchester M13 9PT, UK; te-min.ke@postgrad.manchester.ac.uk (T.-M.K.); artitaya.lophatananon@manchester.ac.uk (A.L.)

**Keywords:** two-sample Mendelian randomization, type 2 diabetes mellitus, pancreatic cancer, FinnGen, UK Biobank

## Abstract

This two-sample Mendelian randomization (MR) study was conducted to investigate the causal associations between type 2 diabetes mellitus (T2DM) and the risk of pancreatic cancer (PaCa), as this causal relationship remains inconclusive in existing MR studies. The selection of instrumental variables for T2DM was based on two genome-wide association study (GWAS) meta-analyses from European cohorts. Summary-level data for PaCa were extracted from the FinnGen and UK Biobank databases. Inverse variance weighted (IVW) and four other robust methods were employed in our MR analysis. Various sensitivity analyses and multivariable MR approaches were also performed to enhance the robustness of our findings. In the IVW and Mendelian Randomization Pleiotropy RESidual Sum and Outlier (MR-PRESSO) analyses, the odds ratios (ORs) for each 1-unit increase in genetically predicted log odds of T2DM were approximately 1.13 for PaCa. The sensitivity tests and multivariable MR supported the causal link between T2DM and PaCa without pleiotropic effects. Therefore, our analyses suggest a causal relationship between T2DM and PaCa, shedding light on the potential pathophysiological mechanisms of T2DM’s impact on PaCa. This finding underscores the importance of T2DM prevention as a strategy to reduce the risk of PaCa.

## 1. Introduction

Pancreatic cancer (PaCa) ranks as the 12th most common cancer and is the 6th leading cause of cancer-related mortality worldwide [1,2]. Although there have been advancements in the treatment of PaCa recently, the 5-year survival rate remains low, at approximately 9% in 2018 [3]. The low survival rate of PaCa can largely be attributed to the majority of diagnoses occurring at locally advanced stages or once metastasis has happened. Diagnosing PaCa at an early stage is challenging, especially in patients who exhibit no specific symptoms. According to Global Cancer Observatory (GLOBOCAN) estimations [1,4], a trend towards an increase in PaCa incidence (+78.3% with 399,571 new cases) and mortality (+81.9% with 382,260 deaths) is projected from 2022 to 2045. Due to the high incidence rate and persistently low survival rate of PaCa, compounded by the absence of a screening program for PaCa, preventing PaCa by the noted risk factors becomes crucial.

Previous literature reviews [3,5,6,7,8,9] have summarized numerous potential risk factors for PaCa, including cigarette smoking, heavy alcohol consumption, obesity, chronic pancreatitis, type 2 diabetes mellitus (T2DM), hepatitis B, cholecystectomy, periodontal disease, aging, male gender, African American ethnicity, non-O blood type, family history, certain hereditary syndromes, and germline mutation. Among these risk factors, T2DM stands out as a growing global health crisis, with an increasingly higher prevalence and disease burden being reported [10,11,12]. In 2017, the global prevalence rate of T2DM was 6059 cases per 100,000 individuals [10]. In our previous UK Biobank cohort study [13,14], a history of diabetes mellitus (DM) was found to at least double the risk of PaCa compared to participants without a DM history [13,14]. Additionally, by calculating the population attributable fraction (PAF), it was revealed that 6% of PaCa cases could be eliminated by avoiding DM [13] in the UK Biobank cohort. Notably, T2DM represents around 90% to 95% of all identified DM cases [15]. Therefore, the ongoing rise in T2DM prevalence worldwide is expected to contribute to an increase in the incidence rate of PaCa.

Despite numerous observational studies [13,15,16,17,18] revealing an association between T2DM and an increased risk of PaCa, the causal relationship between T2DM and PaCa remains controversial in Mendelian randomization (MR) studies. MR studies are important for investigating causal relationships, and T2DM was suggested to be causally associated with PaCa in some two-sample Mendelian randomization (2SMR) studies [19,20]. However, other 2SMR studies [21,22] did not observe a causal relationship between T2DM and the risk of PaCa. The null results may be caused by the limited number of instrumental variables (IVs), which could attenuate the statistical power. On the other hand, in studies observing causal relationships, the potential for overestimation still needs to be considered, as some included SNPs may exert pleiotropic effects on BMI, and some results may be influenced by linkage disequilibrium (LD). Therefore, in this study, we attempt to use comprehensive IVs, avoid SNPs located near genes reported to be associated with BMI, and also check for LD in the IV selection process to enhance the robustness of the results.

MR is one of the epidemiological approaches that adopt genetic variants as IVs to explore causal relationships between exposures and outcomes [23,24,25]. The advantage of MR over traditional observational studies lies in the genetic variants randomly assorted at conception [23,24,25]; however, there are three critical assumptions [26]. These include IVs that are robustly associated with the exposure (relevance assumption [26]), are not linked with any confounders (independence assumption [26]), and influence the outcome solely through the exposure (exclusion restriction assumption [26]). Consequently, MR has the capability to minimize confounding and reduce bias from reverse causation, offering insights comparable to those obtained from randomized controlled trials [23,24,25]. 2SMR is a variation in the MR approach [27,28]. Unlike traditional MR studies, which often require individual-level data on both genetic variants related to the exposure and outcome traits from the same participant dataset, 2SMR can utilize summary-level data from separate genome-wide association studies (GWAS) for both the exposure and the outcome [27,28].

To further explore the underlying mechanisms of PaCa and to elevate awareness of prevention education in public health strategies, understanding the causal relationship between T2DM and PaCa is important. The consistency regarding whether the genetic liability to T2DM is causally related to PaCa remains absent in previous studies [19,20,21,22]. Consequently, we conducted a 2SMR study to comprehensively utilize the most up-to-date GWAS summary data and two large population-based datasets, including FinnGen and UK Biobank, to investigate the associations of genetic liability to T2DM with PaCa.

## 2. Results

### 2.1. Selection of Instrumental Variables

According to the IV selection criteria mentioned in Section 4.2, 414, 423, and 423 SNPs were chosen from the FinnGen, UKBB, and combined FinnGen and UKBB GWAS summary data in the comprehensive model analysis. The details of all selected IVs are provided in Appendix A. In this study, the proportion of variance explained (PVE) or R^2^ by IVs used in the FinnGen, UKBB, and combined FinnGen and UKBB studies were 39.16%, 40.18%, and 40.31%, respectively. The F-statistics for these IVs were 933.97, 958.92, and 961.29. Two SNPs near the *FTO* gene (rs78020297 and rs1421085) were removed in the restricted model analysis.

### 2.2. MR Analysis

In our 2SMR analysis, the inverse variance weighted (IVW) method was designated as the principal method due to its higher statistical efficacy. Additionally, four robust methods, including the Mendelian randomization-Egger (MR-Egger), the weighted median (WM), the weighted mode (WMO), and the Mendelian Randomization Pleiotropy RESidual Sum and Outlier (MR-PRESSO), were employed as complementary approaches to evaluate the genetic causal associations between T2DM and PaCa risk.

In the FinnGen dataset, a causal association between T2DM and PaCa risk was indicated by both the IVW (*p* = 0.033) and MR-PRESSO (*p* = 0.029) methods (Table 1). For a one-unit increase in the log-transformed odds of T2DM, the OR of PaCa risk was estimated at 1.102 (95% CI = 1.008–1.204) by the IVW method and 1.097 (95% CI = 1.010–1.191) by the MR-PRESSO method, respectively (Table 1). The MR-PRESSO global test identified two outlier SNPs; consequently, these outliers were corrected in our MR-PRESSO analysis. After eliminating two SNPs near the *FTO* gene in the restricted model, the causal link between T2DM and PaCa risk remained significant in both the IVW (*p* = 0.046) and MR-PRESSO (*p* = 0.031) methods (Table 1) in the FinnGen dataset. With each unit increment in the log-transformed odds of T2DM, the OR for PaCa risk was 1.095 (95% CI = 1.001–1.198) and 1.094 (95% CI = 1.008–1.187) using the IVW method and MR-PRESSO method, respectively. Five outlier SNPs were detected in the MR-PRESSO global test; therefore, these outliers were removed in our MR-PRESSO analysis.

Within the UKBB dataset, the WM (*p* = 0.022), IVW (*p* = 0.001), and MR-PRESSO (*p* = 0.005) methods revealed a causal relationship between T2DM and PaCa risk (Table 1). These methods indicated a 23.7% increase (OR = 1.237, 95% CI = 1.031–1.482) for WM, an 18.5% increase in the odds of PaCa risk (OR = 1.185, 95% CI = 1.068–1.315) for IVW, and a 16.2% increase (OR = 1.162, 95% CI = 1.048–1.288) for MR-PRESSO, per one-unit increase in the log-transformed odds of T2DM (Table 1). After eliminating two SNPs near the *FTO* gene in the restricted model, the WM (*p* = 0.041), IVW (*p* = 0.002), and MR-PRESSO (*p* = 0.007) methods persistently demonstrated a causal link between T2DM and PaCa risk, as shown in Table 1. A one-unit rise in the log-transformed odds of T2DM correlated with a 23% increase (OR = 1.23, 95% CI = 1.008–1.501) in the WM method, an 18% elevation in PaCa risk odds (OR = 1.179, 95% CI = 1.061–1.310) in the IVW method, and a 15.6% rise (OR = 1.156, 95% CI = 1.042–1.284) in the MR-PRESSO method (Table 1).

For the combined FinnGen and UKBB dataset, the causal effect of T2DM on PaCa risk was demonstrated significantly using the WM (*p* = 0.017), IVW (*p* = 0.001), and MR-PRESSO (*p* < 0.001) approaches (Table 1). With a one-unit increase in the log-transformed odds of T2DM, the OR of PaCa risk was elevated by 15.1% (OR = 1.151, 95% CI = 1.025–1.293) in the WM approach, 13.1% (OR = 1.131, 95% CI = 1.052–1.216) in the IVW method, and 12.7% (OR = 1.127, 95% CI = 1.056–1.204) in the MR-PRESSO method (Table 1). Two outlier SNPs were detected in the MR-PRESSO global test; therefore, these outliers were removed in our MR-PRESSO analysis. After the two SNPs were removed in the restricted model, the IVW (*p* = 0.002) and MR-PRESSO (*p* = 0.001) methods still significantly highlighted the causal effect of T2DM on PaCa risk, as shown in Table 1. For each unit increase in the log-transformed odds of T2DM, there was a 12.5% increase in PaCa risk odds (OR = 1.125, 95% CI = 1.046–1.211) according to the IVW method, and a 12.4% rise (OR = 1.124, 95% CI = 1.053–1.201) using the MR-PRESSO method. The MR-PRESSO global test detected five outlier SNPs; hence, these outliers were eliminated in our MR-PRESSO analysis.

In the scatter plot (Figure 1), the direction of the causal effect of T2DM on PaCa risk was consistently depicted across all MR analysis approaches in the FinnGen, UKBB, and combined FinnGen and UKBB datasets, both in the comprehensive and restricted models.

### 2.3. Sensitivity Analysis

The MR-Egger regression intercept analysis revealed no horizontal pleiotropy in the FinnGen, UKBB, and combined FinnGen and UKBB datasets, both in the comprehensive and restricted models (Appendix A). Furthermore, in our funnel plot visualization (Figure 2), general symmetry suggests the absence of horizontal pleiotropy. In Cochran’s Q test for IVW, heterogeneity (*p* < 0.05) was observed in the analyses of the FinnGen and the combined FinnGen and UKBB datasets (Appendix A). According to the leave-one-out analysis, the sequential removal of each IV did not impact the causal relationship between T2DM and PaCa, nor did it affect the OR. Furthermore, the leave-one-out test did not reveal any potential outliers or evidence of horizontal pleiotropy. The details of the leave-one-out test are listed in Appendix A. In the MR Steiger directionality test, the variance explained in the outcome is less than that in the exposure, confirming the correct causal direction to be true (Appendix A).

### 2.4. Multivariable Mendelian Randomization (MR)

In the multivariable MR analysis, the observed association between T2DM and PaCa remained significant after adjusting for body mass index (BMI) and waist circumference (OR = 1.485, *p* < 0.001, 95% CI = 1.228–1.796) (Table 2).

## 3. Discussion

To explore the causal relationship between T2DM and PaCa, we performed a 2SMR analysis using two large T2DM genome-wide association meta-analyses [29,30] and PaCa cases from the FinnGen and UK Biobank datasets. The IVW method, along with four other robust methods, were utilized in the MR analysis. Sensitivity analyses, the MR Steiger directionality test, and multivariable MR were conducted to strengthen our results. Our findings revealed that genetic liability to T2DM was associated with a higher PaCa risk. Our 2SMR results provided evidence of causal associations between T2DM and PaCa.

The previous two MR studies [21,22] did not observe a causal relationship between genetic liability to T2DM and the risk of PaCa. In the study by Carreras-Torres et al. [21], 44 IVs for T2DM were identified from the previous genetic fine-mapping studies [31,32]. PaCa cases were obtained from the Pancreatic Cancer Cohort Consortium (PanScan) and the Pancreatic Cancer Case-Control Consortium (PanC4). The limited number of IVs, possibly due to the reliance on non-current T2DM GWAS meta-analyses, may have led to reduced statistical power, making it difficult to detect a genuine causal effect. In another study by Chen et al. [22], 231 SNPs associated with T2DM were selected from a genome-wide association meta-analysis [30], with PaCa cases identified from FinnGen, UK Biobank, and PanScan. Despite observing no significant association between genetic liability to T2DM and PaCa in their primary IVW approach, a multivariable MR analysis adjusted for BMI showed that T2DM was associated with a higher OR of 1.19 (95% CI = 1.01–1.40) in the UK Biobank dataset. Given these inconsistent results, further examination is required to investigate whether genetic liability to T2DM is linked to PaCa.

Our results align with two previous 2SMR studies [19,20], indicating a causal association between T2DM and PaCa. In the 2SMR study [19], 83 IVs were selected from the 2018 GWAS meta-analysis [33]. The PaCa cases were obtained from the PanScan and the PanC4. The IVW approach revealed a borderline association between T2DM and PaCa with an OR of 1.09 (95% CI = 1.00–1.19, *p*-value = 0.05) in their restricted MR model. The other 2SMR study [20] used 295 IVs from the DIAGRAM [29] consortium and excluded SNPs in or near the *FTO* gene region. PaCa cases were identified within the UK Biobank. Their results suggested a causal association between genetic liability to T2DM and PaCa, with an OR of 1.12 (95% CI = 1.03–1.21) using IVW methods. An umbrella review [34] that included only three studies [19,20,21] also indicated potential causal associations between genetically predicted T2DM and PaCa. However, the potential for overstated strength and pleiotropic bias in the studies [19,20] still needs to be addressed. For instance, the selected IVs in the prior study [19] included SNPs related to the *FTO* gene, and linkage disequilibrium was not accounted for during the IV selection process in the latter study [20]. In our study, our results are likely to be more robust and well-substantiated by circumventing potential overestimation and pleiotropic effects during the IV selection process.

Our findings from the 2SMR study strengthen the evidence for a causal association between T2DM and PaCa and further support previous observational studies [13,14,17,18,35]. In our two previous UK Biobank cohort studies [13,14], the OR of PaCa was 2.08 and 2.57 in participants with a history of DM compared to controls without a history of DM. Additionally, both new-onset and long-term DM have been reported to approximately double the risk of PaCa [17,18,35]. An umbrella review [34] also revealed a pooled OR of roughly 2 for PaCa risk among patients with T2DM compared to controls.

The pathophysiological mechanisms linking T2DM to PaCa are complex and multifaceted, encompassing insulin resistance and hyperinsulinemia [16,36,37,38], persistent hyperglycemia [39,40,41,42], chronic inflammation [16,43,44], alterations in gut microbiota [45,46,47,48,49], and dysregulated adipokine secretion [16,50,51,52,53,54]. Insulin resistance in T2DM leads to hyperinsulinemia, potentially promoting tumor growth directly by acting on insulin receptors or indirectly by increasing levels of insulin-like growth factor-1 (IGF-1), both of which can stimulate cell proliferation and inhibit apoptosis to enhance cell proliferation pathways [36,37,38]. Additionally, hyperglycemia provides an energy-rich environment for cancer cells, inducing oxidative stress and leading to DNA damage [39,40,41,42]. Concurrently, chronic inflammation associated with T2DM creates a pro-inflammatory environment conducive to pancreatic carcinogenesis [43,44]. Some inflammatory mediators or cytokines, such as interleukin-6 (IL-6) and tumor necrosis factor-alpha (TNF-α), can promote tumor growth and metastasis [43,44]. Changes in the gut microbiota related to T2DM may influence systemic inflammation and metabolic profiles, which may affect cancer development through alterations in bile acid metabolism and the release of metabolites that may have carcinogenic properties [45,46,47,48,49]. Moreover, altered adipokine secretion, characterized by increased leptin and decreased adiponectin levels due to adiposity in T2DM, may facilitate cancer progression through pro-inflammatory and anti-apoptotic effects [50,51,52,53,54]. These interconnected pathways underscore the complex relationship between T2DM and PaCa.

On the other hand, DM may also be a consequence of PaCa [16,55]. For instance, type 3c diabetes (T3cDM), defined as diabetes secondary to pancreatic exocrine disease [16,55], includes cases resulting from PaCa. Therefore, examining the potential for reverse causation is also crucial. In this study, the results of the MR Steiger directionality test verified the correct direction of causality between T2DM and PaCa, thereby enhancing confidence in our causal inference result. Obesity has been widely recognized as a significant risk factor for T2DM [56,57]. To mitigate the confounding bias, SNPs in the *FTO* gene, which are strongly associated with obesity [58], were excluded from the restricted model. Additionally, multivariable MR analyses were conducted with adjustments for BMI and waist circumference. The results from the restricted model 2SMR and multivariable MR robustly support our causal inference, demonstrating that it is not influenced by the confounding factor of obesity.

Recognizing the causal link between T2DM and PaCa can raise awareness about targeting T2DM for pancreatic cancer prevention. Understanding these pathophysiological mechanisms could potentially pave the way for developing targeted prevention and treatment strategies.

This study possesses several strengths. Our 2SMR study design effectively mitigates biases due to confounding, pleiotropy, and reverse causality. Firstly, the IVs for T2DM were obtained from two large European GWAS meta-analyses, enabling us to identify the latest and most comprehensive SNPs related to T2DM and ensuring sufficient statistical power. Additionally, focusing on the European population helped avoid population structure bias in our results. Secondly, the total and average instrument strengths in this study were assessed using the PVE and the F-statistic, which diminished the bias from weak instruments. Furthermore, power calculations for our 2SMR study indicated a robust power estimation. Thirdly, our IVs selection process was high quality and stringent, ensuring all chosen SNPs met the genome-wide significance threshold, were independent of LD, had harmonized strand orientation, and did not include any ambiguous palindromic SNPs. Fourthly, in addition to the main IVW approach, we employed WM, WMO, MR-Egger, and MR-PRESSO as robust methods to minimize potential biases, including pleiotropy, weak instrument bias, and other sources of analytical distortion, thereby enhancing the reliability and validity of the causal inferences drawn from our study. Fifthly, our sensitivity analyses, including the MR-Egger regression intercept test, funnel plots, and leave-one-out test, all confirmed the absence of significant horizontal pleiotropy. Sixthly, the MR Steiger directionality test was performed to ensure the causality direction from T2DM to PaCa was correctly identified. Last but not least, to mitigate the potential pleiotropy associated with obesity, SNPs near the *FTO* gene were excluded from our restricted model. Moreover, multivariable MR analyses were conducted, confirming the causal association between T2DM and PaCa by adjusting for BMI and waist circumference.

Nonetheless, some limitations still need to be considered in this study. Firstly, both the IVs and the outcome data were obtained from European population-based datasets. Hence, this limits our findings from being generalized to other populations. Secondly, heterogeneity among IVs was detected in some FinnGen analyses using Cochran’s Q test for the IVW method. However, the MR-Egger test and other sensitivity analyses did not indicate pleiotropy. In the MR-PRESSO test, causal inference remained significant even after correcting for outliers. Additionally, in the comprehensive model, the WM approach also revealed a significant effect in the combined FinnGen and UK Biobank analysis. Therefore, the likelihood of bias in the results due to pleiotropic effects or invalid IVs is reduced. Lastly, although disease diagnoses are defined by ICD codes and electronic healthcare records, there still exists the possibility of detection bias among T2DM and PaCa cases. Nevertheless, our results are derived from two large GWAS meta-analyses and two extensive population-based datasets, which may overcome individual errors or biases in disease identification.

## 4. Materials and Methods

### 4.1. Study Design

The 2SMR study was used to investigate the causal relationship between T2DM and PaCa. We utilized publicly available summary data from genome-wide association studies (GWAS), including the FinnGen, the UKBB, and two genome-wide association meta-analyses [29,30]. Our research adhered to the three critical assumptions of MR [59]: (1) a strong association between the genetic instruments and T2DM; (2) no association of these instruments with confounding variables; and (3) the exclusive influence of these instruments on PaCa through T2DM.

### 4.2. Genetic Instrument Selection

The selection of IVs for T2DM in this study was based on two genome-wide association meta-analyses [29,30]. The first, known as the DIAGRAM consortium by Mahajan et al. [29], encompassed 74,124 T2DM cases and 824,006 controls of European descent. Second, a genome-wide association meta-analysis by Vujkovic et al. [30] involved 228,499 T2DM cases and 1,178,783 controls of multi-ancestry. The following criteria were applied for the selection of IVs: (1) Initially, we identified single nucleotide polymorphisms (SNPs) from the genome-wide association meta-analyses conducted by Mahajan et al. [29] and Vujkovic et al. [30]. SNPs identified as replication variants were selected for possessing a smaller *p*-value (*n* = 899). (2) SNPs that met the genome-wide statistical significance threshold (*p* < 5 × 10^−8^) were selected as instrumental variables for T2DM (*n* = 589). (3) The clumping threshold for linkage disequilibrium (LD) was set at r^2^ = 0.2 within a 250 kb window [60], using the 1000 Genomes European Panel as the reference. Based on this threshold, 436 SNPs were confirmed as independent. (4) A total of 430 SNPs, 434 SNPs, and 435 SNPs were available in the FinnGen, UKBB, and a combination of FinnGen and UKBB datasets, respectively. (5) To mitigate issues related to the orientation of strands, we flipped the reverse strand to the forward strand, harmonized the effect of SNPs on exposure and outcome, and dealt with the palindromic SNPs. Ambiguous palindromic SNPs that exhibited a minor allele frequency (MAF) greater than 0.42 were discarded [61]. Finally, 414 SNPs in the FinnGen dataset, 423 SNPs in the UKBB dataset, and 423 SNPs in a combined FinnGen and UKBB dataset were selected for comprehensive model analysis. (6) Genetic variants near the *FTO* gene were reported to be associated with BMI [58]. Therefore, to mitigate potential pleiotropic effects, SNPs in the vicinity of the *FTO* gene were excluded from our restricted model analysis (*n* = 412 (FinnGen), *n* = 421 (UKBB), and *n* = 421 (FinnGen + UKBB)). The flowchart of the IV selection is shown in Figure 3.

### 4.3. Outcome Data Source

The GWAS summary data for PaCa used in our study were sourced from the R10 release of the FinnGen Consortium [62,63], including both the FinnGen and UK Biobank GWAS summary data. For detailed information on the web browser, please refer to: https://public-metaresults-fg-ukbb.finngen.fi/ (accessed on 5 January 2024). In the search browser [63], the phenotype “Malignant neoplasm of pancreas (excluding all other cancers in controls)” was used, including 1626 cases and 314,193 controls in FinnGen and 936 cases and 400,294 controls in the UK Biobank. In the FinnGen dataset, PaCa cases were identified using codes from ICD-8, ICD-9, and ICD-10, as well as surgery codes and medication purchase codes. In the UK Biobank, PaCa cases were diagnosed using codes from ICD-9 and ICD-10, surgery records, and self-reported information.

### 4.4. The Strength of the Selection of Instrumental Variables and Power Calculations

To avoid bias from weak instruments in this study, we adopted the proportion of variance explained (PVE) known as R**^2^** for assessing total strength [64] and the F-statistic for measuring average instrument strength [64]. The PVE in the exposure is explained by the selected genetic variants. Generally, a higher PVE is preferable, as it significantly enhances the effectiveness of an MR analysis [64]. The F-statistic was proposed to assess IV strength [65,66]. A commonly used cutoff value is 10 [65,66]; an F-statistic less than 10 indicates weak instruments. The formulas we adopted for the PVE [67] and the F-statistic [67,68] are as follows:(1)$$PVE=2\times EAF\times (1-EAF)\times \beta^{2}$$
(2)$$F-statistic=\;\frac{PVE\times \left(N-K-1\right)}{\left(1-PVE\right)\times \left(K\right)}$$
where β denotes the beta coefficient for the exposure of the SNP according to the GWAS summary data, EAF represents the effect allele frequency of the SNP, *N* refers to the total number of samples, and *K* is the number of IVs.

In this study, the PVE of 436 SNPs after LD clumping was 41.4%, and the F-statistic was 986.35. Both the total and average instrument strengths are considered good.

The power calculation for two-sample MR analysis was conducted using an online tool (https://shiny.cnsgenomics.com/mRnd/ (accessed on 5 January 2024)) [69,70], and the outcomes are presented in Appendix A. The variance explained by the genetic instruments associated with T2DM adopted in FinnGen, UKBB, and combined FinnGen and UKBB studies was 39.16%, 40.18%, and 40.31%, respectively.

### 4.5. Two-Sample Mendelian Randomization (2SMR) Analysis

Numerous MR methods were applied in our 2SMR analysis, including IVW [71] and four other robust methods: the MR-Egger method [72], the WM [73] method, the WMO method [74], and the MR-PRESSO method [75]. The most widely used method is the IVW method, which aggregates Wald estimates of each SNP to derive a comprehensive overall effect estimate [71]. This approach hinges on a crucial prerequisite: the IVs employed in the analysis must be valid and satisfy the three assumptions of MR [71]. Due to its efficiency with valid IVs, the IVW method is often recommended as the primary method for analysis [61]. Nonetheless, to detect potential pleiotropic effects, other robust MR methods should also be performed [61].

The MR-Egger method calculates the causal effect by using the average pleiotropic effect as the intercept [61]. However, its accuracy relies on the indirect effects being unrelated to the exposure, a concept referred to as the Instrument Strength Independent of Direct Effect (InSIDE) [72]. In the WM method, it is assumed that a majority of the variants used are valid instruments (known as the majority valid assumption) [73]. The effect estimate of each genetic variant on the outcome is weighted according to its association with the exposure. The median is then determined. The advantage of the WM method lies in its ability to handle invalid IVs and outliers [61]. Regarding the WMO method, it operates under the assumption that more variants estimate the true causal effect than any other quantity (known as the plurality valid assumption) [61]. Similar to the WM method, weights are assigned to each variant. However, the WMO method utilizes the mode rather than the median. The benefit of the WMO method is its ability to withstand invalid IVs and outliers [74]. MR-PRESSO builds upon the IVW method and includes a global test, an outlier test, and a distortion test [75]. The MR-PRESSO global test is capable of identifying horizontal pleiotropy [61]. If horizontal pleiotropy is detected, it is corrected through outlier removal. Additionally, the distortion test is used to assess whether there are notable disparities in effects before and after the outlier correction process [75].

Hence, in our study, the IVW method was designated as the primary approach, while the other four robust methods were employed as complementary methods. We first performed MR analysis with all the above-selected IVs. If the MR-PRESSO global test identified horizontal pleiotropy, the outliers would be eliminated, and the MR-PRESSO analysis would be repeated. In addition, sensitivity analyses were performed. The MR-Egger regression intercept analysis was conducted to examine horizontal pleiotropy, with a *p*-value < 0.05 considered as evidence of horizontal pleiotropy. Funnel plots were also created for pleiotropy direction detection, where an asymmetrical or skewed pattern may indicate horizontal pleiotropy is present [72]. Furthermore, heterogeneity was assessed through Cochrane’s Q test, where a *p*-value < 0.05 would be considered an indication of heterogeneity. Moreover, a leave-one-out test was executed, systematically removing each SNP to mitigate the potential heterogeneity and consolidate the stability of the estimated causal effect in our study. Ultimately, the MR Steiger directionality test [76] was employed to ascertain the direction of causality by assessing whether the variance explained in the outcome is less than that in the exposure. Our 2SMR analysis was in accordance with the recommendations provided in the STROBE-MR statement [77]. Detailed information is listed in Appendix A. The flowchart of the 2SMR analysis process is shown in Figure 4.

### 4.6. Multivariable Mendelian Randomization (MR)

To mitigate the effects of pleiotropy and reduce bias due to confounding from obesity, we also conducted multivariable Mendelian randomization (MR). Multivariable MR [78] can assess the influence of multiple exposures on the same outcome. Thus, we utilized multivariable MR to explore the causal relationship between T2DM and PaCa by adjusting for BMI and waist circumference. Initially, the 436 T2DM IVs through LD clumping were searched on the PhenoScanner [79,80] website. Of these, 45 SNPs were found to have overlapping traits with “Body Mass Index” and “Waist Circumference”. After harmonizing the SNPs, 43 SNPs were ultimately selected as IVs for our multivariable MR analysis.

### 4.7. Statistical Analysis

A significance level below 0.05 was considered statistically significant, with statistical significance determined by 95% confidence intervals (95% CI) not including one. In this study, the statistical analyses were conducted using R software [81] (version 4.3.2, R Development Core Team, Vienna, Austria). The TwoSampleMR R package [27] and MR-PRESSO R package [82] were employed for all 2SMR analyses, utilizing functions such as harmonise_data, mr, mr_presso, mr_heterogeneity, mr_pleiotropy_test, mr_singlesnp, mr_leaveoneout, mr_scatter_plot, mr_forest_plot, mr_funnel_plot, mv_multiple, and directionality_test [83].

### 4.8. Ethics

All data analyzed in this study were obtained from publicly available GWAS summary datasets. The original GWAS had received approval from the relevant ethics committee. This study did not collect any new data; hence, further ethical approval was not necessary.

## 5. Conclusions

Our findings indicate that a causal relationship between T2DM and PaCa has been established in the 2SMR and multivariable MR studies. This discovery should therefore be used to enhance awareness and the implementation of early prevention and detection strategies for PaCa. These strategies include managing diabetes as a preventive measure against PaCa and emphasizing the importance of controlling blood sugar levels and other metabolic risk factors. Additionally, increasing public awareness of the causal link between T2DM and PaCa could underscore the significance of lifestyle interventions, such as diet habits, physical activity, and weight management, not only for diabetes management but also for reducing the risk of PaCa. Furthermore, understanding the causal pathways could lead to discovering biomarkers and developing pharmacological strategies to improve treatment outcomes and enable more personalized treatment plans. Therefore, further studies are necessary to elucidate the precise pathophysiological mechanisms involved. Ultimately, all these efforts aim to reduce the incidence and mortality associated with PaCa, highlighting the role of T2DM in the development of PaCa and the potential approaches to preventing this disease.

## Figures and Tables

**Figure 1 ijms-25-04615-f001:**
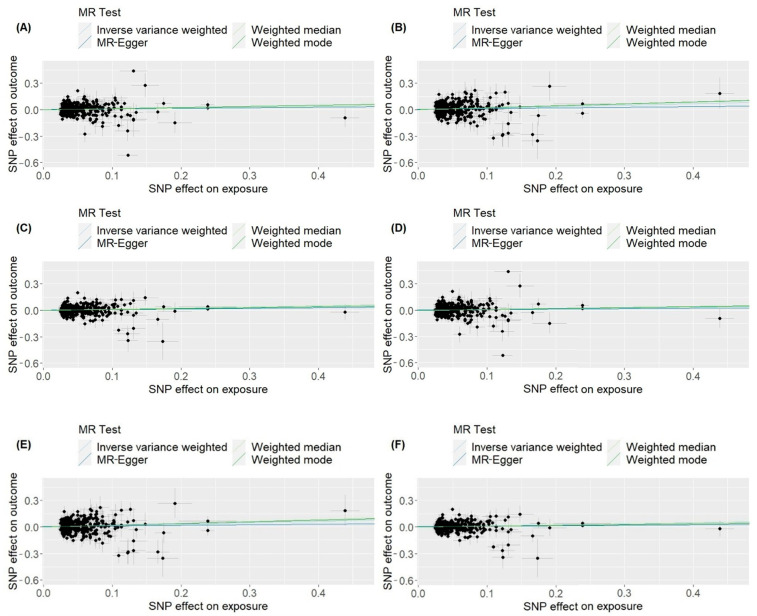
Scatter plots for two-sample Mendelian randomization analyses of the causal effect of type 2 diabetes mellitus on pancreatic cancer. (**A**) Comprehensive model of the FinnGen dataset. (**B**) Comprehensive model of the UK Biobank dataset. (**C**) Comprehensive model of the FinnGen and UK Biobank combined dataset. (**D**) Restricted model of the FinnGen dataset. (**E**) Restricted model of the UK Biobank dataset. (**F**) Restricted model of the FinnGen and UK Biobank combined dataset. MR: Mendelian randomization; SNP: single nucleotide polymorphism; IVW: inverse variance weighted; MR-Egger: Mendelian randomization-Egger; WM: weighted median; WMO: weighted mode.

**Figure 2 ijms-25-04615-f002:**
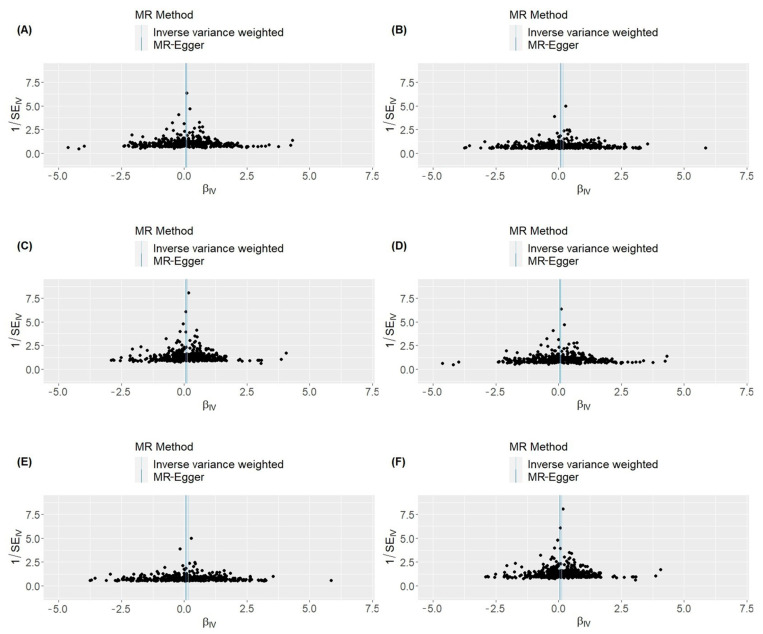
Funnel plot for two-sample Mendelian randomization analyses of the causal effect of type 2 diabetes mellitus on pancreatic cancer. (**A**) Comprehensive model of the FinnGen dataset. (**B**) Comprehensive model of the UK Biobank dataset. (**C**) Comprehensive model of the FinnGen and UK Biobank combined dataset. (**D**) Restricted model of the FinnGen dataset. (**E**) Restricted model of the UK Biobank dataset. (**F**) Restricted model of the FinnGen and UK Biobank combined dataset. MR: Mendelian randomization; βIV: beta coefficient of each instrumental variable, which indicates the estimated effect of each SNP on the exposure variable; 1/SEIV: the inverse of the standard error of the βIV estimation, which indicates the precision or uncertainty of these estimates.

**Figure 3 ijms-25-04615-f003:**
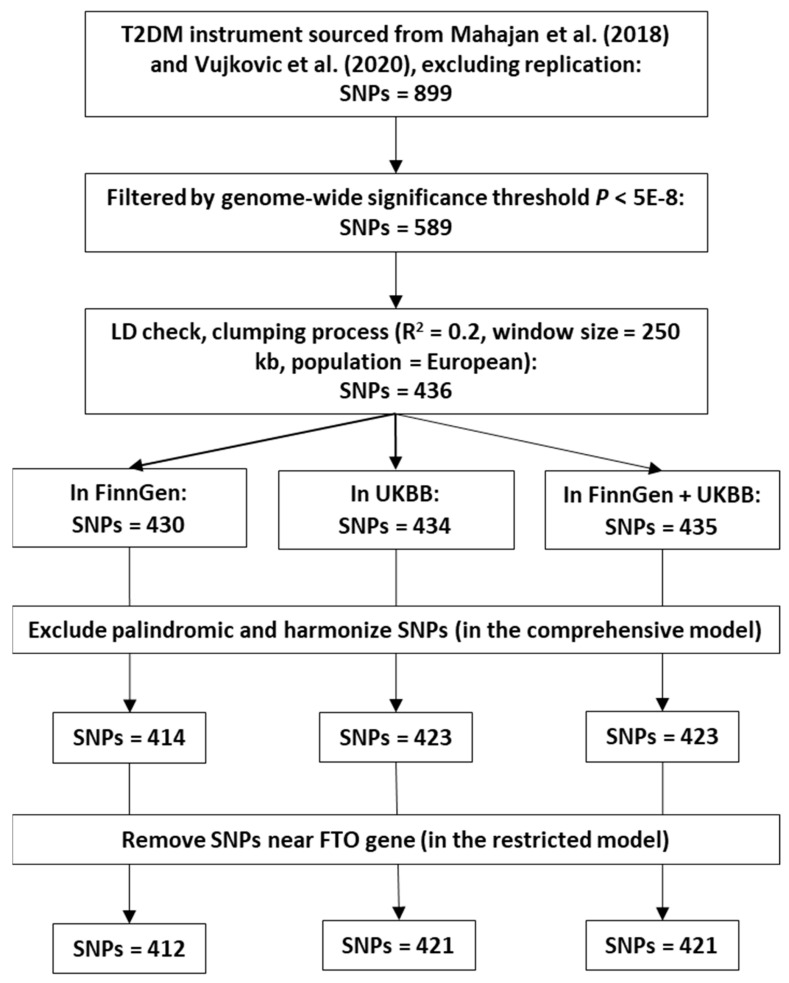
The flowchart for type 2 diabetes mellitus (T2DM) instrument selection. T2DM instrument sourced from Mahajan et al. (2018) [29] and Vujkovic et al. (2020) [30].

**Figure 4 ijms-25-04615-f004:**
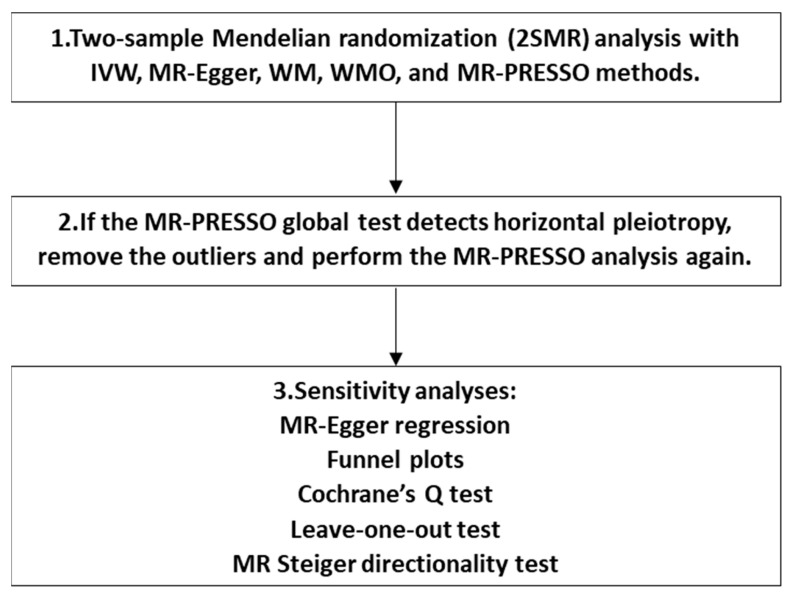
The flowchart of the two-sample Mendelian randomization (2SMR) analysis.

**Table 1 ijms-25-04615-t001:** The two-sample Mendelian randomization (MR) analysis results from the comprehensive and restricted models.

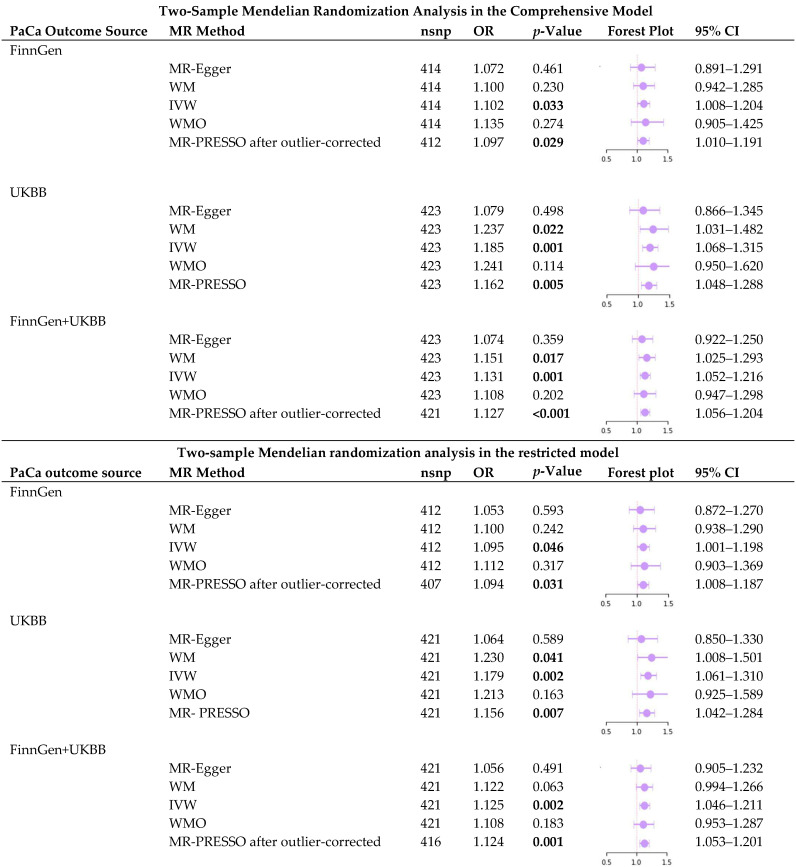

PaCa: pancreatic cancer; MR: Mendelian randomization; nsnp: number of single nucleotide polymorphisms; OR: odds ratio; 95% CI: 95% confidence interval; UKBB: UK Biobank; MR-Egger: Mendelian randomization-Egger; WM: weighted median; IVW: inverse variance weighted; WMO: weighted mode; MR-PRESSO: Mendelian Randomization Pleiotropy RESidual Sum and Outlier. A bold font indicates statistical significance—a *p*-value < 0.05.

**Table 2 ijms-25-04615-t002:** The multivariable Mendelian randomization analysis for type 2 diabetes mellitus adjusted by body mass index and waist circumference.

Multivariable MR					
Exposure	nsnp	OR	*p*-Value	95% Lower CI	95% Upper CI
BMI	43	2.753	0.395	0.267	28.365
T2DM	43	1.485	**<0.001**	1.228	1.796
Waist circumference	43	0.190	0.266	0.010	3.545

MR: Mendelian randomization; nsnp: number of single nucleotide polymorphisms; OR: odds ratio; CI: confidence interval; BMI: body mass index; T2DM: type 2 diabetes mellitus. A bold font indicates statistical significance—a *p*-value < 0.05.

## Data Availability

The datasets analyzed during the current study are available in the FinnGen repository (https://r10.finngen.fi/ (accessed on 5 January 2024)). The summary statistics can be accessed by applying at: https://elomake.helsinki.fi/lomakkeet/124935/lomake.html (accessed on 5 January 2024).

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
