# Peer review of "Strengthening the Evidence for a Causal Link between Type 2 Diabetes Mellitus and Pancreatic Cancer: Insights from Two-Sample and Multivariable Mendelian Randomization"

_ijms, 2024, doi:10.3390/ijms25094615_

Round 1
Reviewer 1 Report
Comments and Suggestions for Authors
This manuscript by Ke et al addresses an important question of whether T2D has a causal effect on the development of PaCa, a rare cancer with poor prognosis. Using Mendelian randomization their study builds on previous investigations that have shown no or limited association, providing robust evidence for a causal effect. A wide range of methods have been used that validate and strengthen their findings. Their main findings are supported by secondary MR methods, and across 2 cohort populations (UKB, FinnGen, and combined). The findings are unchanged when excluding SNPs at the BMI-associated FTO locus, as well as in multivariate MR analysis including BMI as a covariate. The findings are clearly presented and discussed, with well-described background, methodology and strengths and limitations.
I have only minor questions and comments.
Introduction- the authors appropriately raise that previous studies have yielded inconsistent results. While the potential reasons for this are well discussed in the discussion, the novelty of the current study and how it aims to overcome the previous inconsistencies could be made clearer in the Introduction.
Methods- The authors use two large population cohorts UKB and FinnGen in their analyses, and although PaCa is a rare disease they were well powered enough to detect a robust effect. That said, can the authors comment on whether they considered also including pancreatic cancer consortia data such as PanScan in the analyses, to use everything currently available/accessible to provide the best possible estimate?
Figure1 scatter plot text is difficult to read
The final sentence of the Conclusion at line 456-7 could be clearer.
There is also some repeated text at lines 331-322.
Small terminology issues- it should be diabetes mellitis rather than diabetic mellitis, and sometimes FinnGen is inappropriately shortened to "Finn". Inverse variance weighting -> weighted. The capitalisation in "Mendelian randomization pleiotropy Residual Sum and Outlier" doesn't seem to match the acronym MR-PRESSO.
Reviewer 2 Report
Comments and Suggestions for Authors
The manuscript, 'Strengthening the Evidence for a Causal Link between Type 2 Diabetes Mellitus and Pancreatic Cancer: Insights from Two-3 Sample and Multivariable Mendelian Randomization' suggests a genetic link between T2DM and pancreatic cancer.
The study was well conducted and used different MR methods, which contributed to the casual association. The study is clearly formulated, and analyses were adjusted for confounding factors such as BMI and waist circumference.
There are some minor suggestions for the authors:
In lines 331-332, this sentence fragment is written twice: 'Genetic variants near the FTO gene.'
Figure 1—scatter plots and Figure 2 - funnel plots = The figure legends need to be written in the larger font; they are now illegible.
Comments on the Quality of English LanguageMinor editing of the English language is required.
